# A Study of the Friction Characteristics of Rubber Thermo-Mechanical Coupling

**DOI:** 10.3390/polym16050596

**Published:** 2024-02-21

**Authors:** Junyu Liu, Meng Wang, Haishan Yin

**Affiliations:** National Engineering Laboratory of Tire Advanced Equipment and Key Materials, Qingdao University of Science and Technology, Qingdao 266100, Chinaardbeg@163.com (M.W.)

**Keywords:** friction model, thermo-mechanical coupling, tire rubber, rubber friction

## Abstract

The friction performance of tread rubber is related to the safety of the vehicle during driving, especially in terms of shifting speeds, cornering, and changing environmental factors. The experimental design used in this paper employed a self-developed automatic multi-working-condition friction tester to investigate the correlation between the friction coefficient of three tread formulations and various factors, including speed, pressure, temperature, side deflection angle, and lateral camber. This experimental study demonstrates that the coefficient of friction decreases with increasing load and increases with increasing sliding velocities due to changes in adhesion friction. Due to the increasing and decreasing changes in rubber adhesion and hysteresis friction caused by temperature, the coefficient of friction shows a tendency to increase and then decrease with the increase in temperature; thus, temperature has an important effect on the coefficient of friction. Based on the basic theory of friction and experimental research, the Dorsch friction model was modified in terms of temperature, and the analytical relationship between the rubber friction coefficient and the combined variables of contact pressure, slip velocity, and temperature was established, which is more in line with the actual situation of rubber friction. The model predictions were compared with the experimental results, and the error accuracy was controlled within 5%. This verifies the accuracy of the model and provides a theoretical basis for the study of rubber friction.

## 1. Introduction

Tires play a critical role in ensuring safe automobile handling. The force and torque required for driving are generated by the friction between the tire and the road. Therefore, the tire’s friction performance directly affects the vehicle’s handling performance, which, in turn, impacts the safety of the passengers and their vehicle. Rubber is the primary component of a tire, and its dynamic friction characteristics are influenced by a number of factors, such as speed, pressure, temperature, sideways deflection angle, and sideways inclination angle, with temperature in particular having a significant effect on these characteristics [1,2,3]. During driving, tires experience high-frequency deformation due to the hysteresis heat and friction with the ground. This causes the tire temperature to increase, and when it approaches the critical temperature, the physical properties of the tire decrease significantly. In certain conditions, this can lead to tire delamination or even blowout, posing a safety hazard to the vehicle and its occupants.

Due to rubber being viscoelastic, meaning that its friction properties are sensitive to temperature, an increase in temperature affects the shear modulus and coefficient of friction of the rubber, leading to changes in the mechanical properties of tires [4,5]. During actual operation, tires are affected by high-frequency alternating loads and hysteresis heat generation, exhibiting typical thermal coupling characteristics.

It is generally believed that the friction process between the tire and the road can be divided into two parts. The first is the road surface and the rubber surface molecules due to intermolecular forces, leading to adhesion and fracture, constituting adhesion friction. The second is hysteresis friction, constituting the sticky loss of rubber due to external forces [6], as shown in Figure 1.

When the rubber material slides along the road surface, the surface molecules of the rubber and the road surface molecules adhere under the influence of intermolecular forces. This adhesion elongates the molecules in the sliding direction, leading to fracture and affecting the relative movement between the rubber and the pavement. The mechanism of adhesion involves the formation, destruction, and re-formation of joints on both sides of the material and the contact surface. The energy lost due to adhesion during sliding is numerically equivalent to the work performed by adhesion friction.

When rubber materials are in contact with hard objects, the phenomenon of large elastic deformation and recovery deformation will occur repeatedly in the process of movement due to their viscoelasticity. When rubber molecules come into contact with the micro-convex bodies on the road surface, the energy generated by the compression of the rubber as it approaches the micro-convex body is greater than the energy recovered after crossing the micro-convex body due to hysteresis loss. This will produce two peaks, as shown in Figure 1. Its horizontal component is the hysteresis friction force. The energy lost due to the force of hysteresis friction during sliding is numerically equivalent to the work performed by hysteresis friction.

Schallamach [7] proposed that rubber has properties similar to viscous fluids and studied the relationship between the sliding speed and the coefficient of friction. The experiments showed that the coefficient of friction of rubber gradually increases with the sliding speed, reaching a peak at a particular speed, which then gradually decreases. Schallamach pointed out that the decrease in the coefficient of friction is due to the increase in temperature during the friction process. Savkoor [8] concluded that the coefficient of friction under steady-state conditions is primarily influenced by adhesion friction, which is dependent on the sliding speed and temperature under normal pressure. Dorsch [9] measured the variation in the coefficient of friction of tread rubber with sliding speed and contact pressure when the tire was rolling in a sideways bias. Dorsch considered speed, contact pressure, and temperature to be the three parameters that affected the coefficient of friction, but modeling temperature, which is relatively difficult to control and measure, was discarded. Canudas-de-Wit C et al. [10] used the LuGre model to describe the friction characteristics between the tread and the ground of a vehicle under braking conditions and tested the accuracy of the model. Liang K [11] considered the influence of friction heating and established an improved LuGre model in order to better capture the change in friction torque during multiple in situ turns caused by frictional heating. Shao R [12] devised a dynamic friction model for rubber considering the dissipative elasticity and adaptive transient properties of bristle models. This model can describe the stick–slip process of the rubber surface and the relaxation of the friction force at a standstill and its stick–slip effects. At present, research on the friction mechanism and model between tires and road surfaces is insufficient to accurately characterize the relationship between the friction coefficient and the influencing factors. This is particularly true for the thermal coupling characteristics of rubber friction.

In this regard, it is necessary and urgent to develop a fully automatic multi-working condition friction tester to study the correlation between rubber friction characteristics and various factors such as speed, pressure, temperature, side deflection angle, and side inclination angle and to establish the characterization of rubber friction including multiple factors.

## 2. Experimental Section

### 2.1. Test Samples

In this research, three tire-tread formulations with large differences in performance were used for the experiments, formula 1# used natural rubber and N234 carbon black, which is suitable for paved roads, formula 2# used No. 3 smoke rubber and N115 carbon black, which is suitable for engineering transportation, and formula 3# used STR20, polybutadiene rubber, and N234 carbon black, which is suitable for high-speed road surfaces. The main materials used in the preparation of the test were natural rubber, carbon black, zinc oxide, sulfur, stearic acid, and antioxidants, etc. The types of materials and ratios of the three formulations are shown in Table 1. NR and BR is produced in Vietnam. 3#RSS and STR20 is produced in Thailand. Carbon black and white carbon black are produced in Cabot Tianjin, China. The remaining additives are commercially available.

### 2.2. Test Equipment and Methods

#### 2.2.1. Experimental Equipment

The automatic multi-working condition friction testing machine developed in this work can measure the rubber friction coefficient under multiple variables. The variable parameters included speed, pressure, temperature, side deflection angle and side inclination angle, abrasive surface, etc.

Currently, there are several types of rubber wear testing machines in use. Din, Williams, and Akron wear machines are commonly used in simple conditions, while LAT100 and Lambourn wear machines are used in more complex conditions, with their contact forms and friction mechanisms shown in Figure 2a, Figure 2b, Figure 2c, and Figure 2d, respectively. However, these abrasive machines cannot simulate the complex conditions that the tire experiences while in use, such as the environment, road conditions, and running posture. As a result, the laboratory test results may be inconsistent with tire road test results or even contradictory. Conversely, road experiments with actual vehicles may present challenges, such as long cycles, high costs, the significant influence of driving habits, and poor data controllability. To accurately simulate the thermodynamic behavior of tire wear under complex operating conditions, this machine was developed. Its main characteristics are as follows.

① Contact pattern: The contact form of this equipment is the friction contact between the rubber wheel and the plane friction belt as shown in Figure 2e, which is different from the circular friction vice of (a) and (d) and the rotary friction vice of (b) and (c); it can better simulate the friction contact form between the tire and the road surface. The road simulation device adopts the track type, which is in planar contact with the rubber wheel and closer to the actual road surface contact state of the tire. The track is driven by a combination of multiple wheels, and the top wheel can be lifted and moved to control the track tension and facilitate replacement. The track surface can be designed with sand belts of differing sharpness and roughness to simulate the actual road surface.

② Operating attitude: It integrates various operating attitudes of the rubber wheel and can guide the setting of inputs, including the operating attitude of the tire (acceleration, braking, lateral rolling, load), environmental factors (temperature, humidity, road conditions), and other complex operating conditions. The functional area of the main machine of the fully automatic multi-working condition friction tester is shown in Figure 3. The flat structure wheel ensures that the friction between the rubber wheel and the sand belt is the friction between the circle and the plane. Both the linear speed of the rubber wheel and the belt are adjusted in the rubber wheel drive area and the belt drive area. The rubber wheel deflection and tilt function area allows for changes in the rubber wheel’s running attitude and helps to keep it stable during the experiment. The pressurizing machine provides transverse pressure to the rubber wheel, which is necessary to complete the loading function. This is achieved by driving the friction belt fixing plate to exert pressure on the rubber wheel.

③ Temperature control: Rubber wheel heating is accomplished by connecting the conductive slip ring to the heating jacket through the heating wire. The heating jacket is used to heat the rubber wheel internally, as shown in Figure 4. The infrared thermometer can detect the average temperature of the friction surface in real time, allowing precise temperature control of the rubber wheel.

④ Special rubber wheels: The test sample needs to be made into a rubber wheel, which consists of a metal sun wheel on the inside and a rubber layer covering the outside. The tooth-shaped groove on the surface can prevent the outer rubber layer sliding and falling off of the metal sun wheel during the friction experiment. The outer diameter of the rubber wheel is 90 mm, the inner diameter is 40 mm, and the thickness is 30 mm, as shown in Figure 5a. Rubber wheels are made from matching vulcanizing molds, as shown in Figure 5b.

#### 2.2.2. Experimental Methods

(1)Production of the rubber wheel sample

The process included adopting the tread rubber formula, using the plasticizing SXM-500 refining machine, composing the mixed rubber, placing the mixed rubber into the mold (Figure 5c), and conducting flat vulcanization at 150 degrees for 25 min to prepare the test rubber wheel. The surface of the rubber wheel should be flat, without impurities, cracks and depressions, and other defects.

(2)Pretreatment of the rubber wheels

We used the automatic multi-working condition friction tester for abrasion pretreatment of the rubber wheel, and we set the linear speed of the rubber wheel to 0.8 m/s, the linear speed of the abrasive belt to 0.6 m/s, the formation of the linear velocity difference to 0.2 m/s, the contact pressure to 200 N, and the deflection angle and the tilt angle to 0°. The temperature was room temperature, and the running time was 4 min. The rubber wheel pre-grinding aims to eliminate the defects on the surface of the specimen so that the sample friction force is more uniform. The surface debris of the rubber wheel was cleaned, and the wheel was kept at room temperature for more than 20 min to eliminate the influence of the temperature rise of the rubber wheel caused by pre-grinding. This ensured that the initial temperature of the rubber wheel met the requirement of 25 °C.

(3)Friction test

The rubber wheel was installed and fixed in the friction chamber, as shown in Figure 3. According to the experimental requirements, in the computer interface, in order to achieve the correct rubber wheel temperature, rubber wheel linear speed, abrasive belt linear speed, contact pressure, tilt angle, deflection angle, experimental time, and other parameters of the settings, after the start button was pressed, the test machine automatically ran according to the set parameters and automatically stopped at the end of the test period. Each rubber wheel was tested 5 times, and the data were collected after the data stabilized. We checked the wear condition of the rubber wheel after each test to eliminate the influence of the previous test on the following test data.

(4)Analysis of test results

At the end of the test, the computer derived the experimental conditions of each parameter set and the corresponding lateral force, longitudinal force, and moment test results. We carried out three reproducibility experiments for each condition to remove the fluctuating test results and took the mean value of the more stable experimental results for the calculation and analysis of the friction coefficients.

### 2.3. Selection of Experimental Conditions

Rubber friction properties are influenced by various factors, including the rubber formulation, speed, pressure, and temperature. In this study, we selected three tread formulations, and combined tests were conducted on factor variables such as temperature, line speed difference, pressure, and deflection angle to investigate the effects of different factors on the rubber friction properties. The experimental conditions are shown in Table 2. This experiment mainly studied the friction performance of tire rubber under normal-temperature and high-temperature driving conditions on dry and rough road surfaces, simulated the most common driving conditions of cars in daily life, and did not set conditions near the glass transition temperature of rubber; hence, relevant experiments should be further carried out.

The friction vice material comprised a sand belt of zirconium corundum material, with a mesh number of 80 mesh. The characteristics of the rubber wheel sliding contact with the road surface are very important, and the experimental results produced by different roads may be completely different; for example, the friction coefficient results of the glass road surface and the silicon carbide sand belt experiment tend to be completely different [2]. This can be discussed in a follow-up study.

## 3. Experimental Results and Discussion

### 3.1. Effect of the Contact Pressure on the Friction Properties

Friction experiments were carried out on the rubber wheels prepared in the three formulations under different contact pressures and different temperature conditions, in which the linear speed of the abrasive belt was 500 mm/s, the linear speed of the rubber wheel was 800 mm/s, the deflection angle was 0°, and the friction test was carried out on the three formulations at the contact pressures of 100, 200, 300, and 400 N and the temperatures of 25, 40, 55, 70, and 85 °C. The study obtained the trend of the variation in the friction coefficient for the three formulations at different contact pressures and temperatures, as shown in Figure 6. The experimental data are shown in Appendix A and Appendix B.

According to Figure 6, the trend of the friction coefficient with the contact pressure can be seen; the friction coefficient of the three formulations is correlated with the pressure. The friction coefficient of formula 2# decreases from 0.65 to 0.48 when the pressure is increased from 100 N to 200 N at a temperature of 25 °C, and the friction coefficient decreases to 0.35 when the contact pressure continues to be increased to 400 N. The friction coefficient of formula 1# decreases from 0.96 to 0.78 when the pressure is increased from 100 N to 200 N, the friction coefficient of formula 1# decreases from 0.96 to 0.78 at a temperature of 40 °C, and the friction coefficient decreases to 0.60 when the contact pressure continues to be increased to 400 N. The coefficient of friction decreased to 0.60 at 400 N.

The coefficient of friction decreases with the increase in the load, which shows that the increased speed of the friction force is less than that of the normal force, and the relationship is nonlinear. Gaetano Fortunato [13] explored the reasons for several of these aspects and found that the main reason was friction heating, which softened the rubber, increased the contact area, and in most cases reduced the contribution of the viscoelasticity to the friction. Andrej Lang [14] studied the dry friction properties of tire tread rubber in contact with rough granite. It was pointed out that the strong load dependence of the coefficient of friction at high temperatures (40 °C to 100 °C) was mainly due to the contribution of adhesion. Y. Fukahori [15] changed the friction coefficient from the adhesion term and hysteresis term to the adhesion term, deformation term, and crack generation term. It is pointed out that when the normal load increases, the contribution of the deformation term and crack generation term increases slightly, but the adhesion term decreases more obviously, resulting in a significant decrease in the total friction coefficient. The data from this experiment are very similar to previous studies.

It can be seen that at the same contact pressure, the difference in temperature leads to different friction coefficients as well; this phenomenon is more pronounced at lower pressures, and the study of friction modeling cannot ignore the effect of temperature.

### 3.2. Effect of the Slip Velocity on the Friction Properties

Friction experiments were carried out on the rubber wheels prepared in the three formulations under different slip velocity and temperature conditions. The linear velocity difference refers to the difference in velocity between the rubber wheels and abrasive belts, also known as the slip velocity. The contact pressure was 200 N, the deflection angle was 0°, and the friction test was carried out for the three formulations when the line speed difference was 0, 100, 200, 300, and 400 mm/s, and the temperature was 25, 40, 55, 70, and 85 °C. The study obtained the trend of the variation in the friction coefficient for the three formulations at different contact pressures and temperatures, as shown in Figure 7. The experimental data are shown in Appendix A and Appendix B.

According to Figure 7, the trend of the friction coefficient with the slip speed can be seen, and there is a correlation between the friction coefficient of the three formulations and the slip speed. Fukahori [15] suggested that the adhesive friction on rubber surfaces increases almost proportionally with the increase in velocity. Fucheng Guo [16] obtained similar results using molecular dynamics. As the shear velocity increases, the friction coefficient first increases and then decreases. The change in the friction coefficient with the velocity can be attributed to the change in the average longitudinal stress during friction.

The friction coefficient between the rubber wheel and the abrasive belt gradually increased as the slip speed increased with the increase in the line speed difference in the experimental range. It is worth noting that the surface temperature was monitored in real time, and its value was controlled to be basically unchanged when the variable test of slip speed was carried out. Thus, the coefficient of friction increased as the slip speed increased, and the impact of the temperature rise due to the slip friction on the coefficient of friction could be disregarded. Here, the effect of the temperature on the friction coefficient was achieved by the heating method inside the rubber wheel. As can be seen from Figure 7, at the same slip speed, the difference in temperature resulted in a substantial change in the friction coefficient as well.

### 3.3. Effect of the Sideslip Angle on the Friction Properties

The deflection angle is the angle between the center surface of the rubber wheel and the forward direction of the rubber wheel. The linear speed of the abrasive belt was 0.5 m/s, the linear speed of the rubber wheel was 0.8 m/s, the contact pressure was 200 N, and the friction test was carried out on the three formulations at the temperatures of 25, 40, 55, 70, and 85 °C and the deflection angles of 0, 10, 20, and 30°. The trends in the friction coefficients of the three formulations at different deflection angles and temperatures are shown in Figure 8. The experimental data are shown in Appendix A and Appendix B.

According to Figure 8, the trend of the friction coefficient with the change in the sideslip angle shows that the friction coefficient of rubber is correlated with the sideslip angle. The coefficient of friction between the rubber wheel and the abrasive belt increased gradually with the increase in the deflection angle in the experimental range. This is in good agreement with the corresponding increase in the coefficient of friction as the difference in line speed increased. As can be seen from Figure 8, at the same deflection angle, the difference in temperature led to a large difference in the coefficient of friction. Temperature has a strong effect on the friction coefficient under different pressure conditions, linear velocity differences, and deflection angles. It can be seen that the friction characteristics of rubber have typical thermal–mechanical coupling characteristics.

### 3.4. Effect of the Temperature on the Friction Properties

According to Figure 6 (data are shown in Appendix A and Appendix B), the trend in the friction coefficient between the temperature and different contact pressures can be obtained, as shown in Figure 9.

According to Figure 7 (data are shown in Appendix A and Appendix B), the trend in the friction coefficient between the temperature and different slip speeds can be obtained, as shown in Figure 10.

According to Figure 8 (data are shown in Appendix A and Appendix B), the trend in the friction coefficient between the temperature and different deflection angles can be obtained, as shown in Figure 11.

The trend in the area of the pressure impression of rubber wheels at different temperatures can be seen in Figure 12.

According to Figure 9, Figure 10 and Figure 11, the trend of the friction coefficient with the temperature can be obtained. It is obvious that the friction coefficients of all three formulations are closely related to the temperature. In the experimental range, with the increase in the temperature, the friction coefficient of the three formulations showed the distribution of first increasing and then decreasing, and the high point of the friction coefficient appeared at about 70 °C. The reason for this is that the increase in temperature makes the thermal movement of rubber molecules more intense, resulting in a decrease in the rigidity of the rubber wheel and a decrease in the hysteresis loss. Reducing the rigidity of the rubber wheel leads to an increase in the contact area between the rubber wheel and the abrasive belt, the average pressure decreases, the adhesion friction increases, and the friction coefficient shows an increasing trend. Based on Figure 13, it can be seen that after the temperature increases to a certain degree (about 55 °C), the rubber wheel deformation slowed down and almost no longer increased; that is, the adhesion friction no longer increased. The hysteresis loss of the rubber wheel gradually decreased with the increase in the temperature, and when the reduction in the hysteresis friction exceeded the increment in the adhesion friction (about 70 °C), the friction coefficient showed a decreasing trend. This results in the friction coefficient showing a trend of first increasing and then decreasing with the increase in the temperature, which is related to the temperature change leading to the increase and decrease in the adhesion friction and hysteresis friction conversion.

In addition, the continuous increase in temperature will cause shear stress, tensile stress, relaxation stress reduction, and rubber surface hardness, resulting in a substantial decline in the physical and mechanical properties of the rubber [17], while the high temperature in the rubber surface will intensify the chemical reaction between the rubber macromolecules and oxygen, causing thermal degradation and re-crosslinking of the rubber [18]. High temperatures have a larger negative impact on rubber fatigue, abrasion friction, and other comprehensive properties.

## 4. Friction Model Development and Validation

### 4.1. Modeling Friction

Dorsch [9] conducted friction tests on rubber wheels under different loading conditions, speeds, and temperatures and concluded that the sliding speed, contact pressure, and temperature were the three parameters that independently affected the friction coefficient and that the friction coefficient is a nonlinear function of the contact pressure, sliding speed, and temperature. In order to facilitate engineering applications, Dorsch discarded the relatively difficult control and measurement of the temperature parameters in the model establishment, based on experimental data, to establish a friction model parameterized by contact pressure *p* and sliding velocity *v*. The corresponding characterization equation is:(1)$$\mu \left(p,v\right)=c_{1}p^{c_{2}}v^{c_{3}}\;,$$
where *c*_1_, *c*_2_, and *c*_3_ are friction-related coefficients. Alternatively, a quadratic polynomial can be used to express the corresponding characterization equation as:(2)$$\mu \left(p,v\right)=c_{0}p+c_{1}p^{2}+c_{2}v+c_{3}p^{2}+c_{4}pv+c_{5}v^{2}\;,$$
where *c*_0_, *c*_1_, *c*_2_, *c*_3_, *c*_4_, and *c*_5_ are the coefficients of the friction correlation, *v* is the velocity, and *p* is the contact pressure. The friction correlation coefficients, *c*_0_, *c*_1_, *c*_2_, *c*_3_, *c*_4_, and *c_5_*, are related to the pavement roughness, grain size, and rubber formulation.

With the development and maturation of computer technology and numerical simulation technology, the temperature parameters can be obtained through the numerical simulation of rubber products; using the Dorsch model, we established the friction coefficient and the pressure and slip velocity power function based on the relationship between the temperature and the coefficient of friction, further studied the temperature and friction coefficient of the relationship, and established the rubber friction coefficient of the thermodynamic coupling model. Additionally, the rubber products’ (such as tires, etc.) thermodynamic numerical simulation technology, used to achieve engineering applications, has very important theoretical value and practical significance.

Early researchers studied the effect of the temperature on the friction coefficient using pure rubber at low speeds and smooth surface conditions. They discovered a bell-shaped relationship between the coefficient of friction and $\log a_{t}V$ through the WLF equation [19]. In recent years, Tolpekina [20] studied the relationship between the friction coefficient and log⁡V by using tread rubber with carbon black filler, and the pattern was slightly different. The research on tread rubber, especially its applicability under complex conditions, needs to be further explored. Tires are usually operated on rough roads and at high speeds, and in order to increase the strength and abrasion resistance of tires, tread rubber is usually filled with a large amount of carbon black. The temperature of the tread can reach around 80 °C due to hysteresis heat generation and friction with the ground during high-speed running. Therefore, studying the effect of the temperature on the friction is crucial. Therefore, the temperature variable is taken separately for research, to exclude the influence caused by other factors.

In investigating the relationship between the friction coefficient and temperature, it can be seen from Figure 9 and Figure 10 that the relationship between the friction coefficient and the temperature presented a better regularity, and the highest point of the friction coefficient occurred at about 70 °C. The whole curve showed a trend of bell-shaped distribution; formulas 1#, 2#, and 3# presented the same curve characteristics, but the curves under the conditions of different formulas and factors presented certain differences, which was manifested in the different slopes of the curves, and the initial intercept was also different.

Dorsch suggests that localized molecules between the rubber and the road surface adhere due to intermolecular forces, and during sliding, the adhesive portion is stretched to the point of fracture and then relaxed. Adhesion formation and fracture are both thermal activation rate processes. Schallamach [7] made a similar point about the viscous fluid’s relationship between fluidity and temperature during flow. M. Kluppel [21] carried out the construction of viscoelastic master curves and showed that the network of fillers in filled rubber dominates their dynamic mechanical properties, exhibiting Arrhenius-dependent properties. This process of thermal activation rate can be described by the Arrhenius equation:(3)$$k=Ae^{\frac{\Delta E}{RT}}\;,$$
where *A* is the pre-exponential factor, *E* is the apparent activation energy, *R* is the molar gas constant, *T* is the thermodynamic temperature, and *k* is the rate coefficient.

The magnitude of the rate coefficient *k* in the Arrhenius equation reflects the rapidity of the rate of adhesion formation and fracture, and the friction essentially includes the adhesion and fracture of molecules; so, the relationship between the friction and temperature is similar to the relationship between the rate coefficient and the thermodynamic temperature in the Arrhenius equation, i.e., the model of the exponential function with the base *e*. Combined with the curve characterization in Figure 9, the established friction coefficient versus temperature is characterized as:(4)$$\mu \left(t\right)=\alpha e^{-\beta {\left(t-t_{m}\right)}^{2}}+\gamma \;,$$
where *γ* denotes the intercept, *t_m_* denotes the temperature corresponding to the maximum coefficient of friction, and *α* and *β* characterize the curve slope characteristic correlation coefficients related to the rubber’s viscoelasticity.

The characterization of the friction coefficient for speed and pressure is constructed according to the Dorsch model as follows:(5)$$\mu \left(p,v\right)=a_{1}p^{b_{1}}v^{b_{2}}+a_{2}\;,$$
where *a*_1_, *a*_2_, *b*_1_, and *b*_2_ are friction-related coefficients.

According to Equations (4) and (5), the characterization formula of the thermodynamic coupling comprehensive friction model is established as follows:(6)$$\mu \left(t,p,v\right)=\mu \left(t\right)\cdot \mu \left(p,v\right)=\left(\alpha e^{-\beta {\left(t-t_{m}\right)}^{2}}+\gamma \right)\left(a_{1}p^{b_{1}}v^{b_{2}}+a_{2}\right)\;.$$

It is worth noting that *a*_1_, *a*_2_, *b*_1_, and *b*_2_ in $\mu \left(p,v\right)$ are the parameters obtained by fitting using the test data at 25 °C, the relationship between the friction coefficient and the temperature $\mu \left(t\right)$, is investigated with $t_{0}$ as the initial temperature, and variable substitution needs to be carried out using $\mu \left(t\right)=\mu \left(T-t_{0}\right)$ to eliminate the effect of the initial temperature, where $t_{0}$ = 25 °C.

In the equation, *α* and *β are the* rubber viscoelastic correlation coefficients, *γ* is the temperature correlation coefficient, tm is the temperature corresponding to the time when *μ* takes its maximum value, *a*_1_ and *a*_2_ are the correlation coefficients for the ground shape and pavement roughness, *b*_1_ is the correlation coefficient for the sensitivity of the rubber to the pressure, *b*_2_ is the correlation coefficient for the sensitivity of the rubber to the slip velocity, *t* is the temperature, °C, *p* is the contact pressure, *N*, and *v* is the slip velocity, mm/s.

The data obtained from the experiments were divided into two parts, shown in Appendix A and Appendix B. The data in Appendix A were used to fit the relevant parameters of the model, and the data in Appendix B were used to validate the accuracy of the model. The friction coefficients of the three tread adhesives were fitted by MATLAB using nonlinear least squares and multiple linear regression according to Equation (6). The experimental results of the contact pressure change at 25 °C for formula 1#, shown in Appendix A, were selected for curve fitting to obtain parameter *b*_1_. The experimental results for the change in the slip velocity at a temperature of 25 °C for formula 1#, shown in Appendix A, were selected for curve fitting to obtain parameters *a*_1_, *b*_2_, and *a*_2_. Then, the experimental results of the temperature change of formula 1#, shown in Appendix A, were selected for curve fitting to obtain the parameters *α*, *β,* and *γ*. The correlation coefficients of formula 1# were obtained, and the correlation coefficients of the integrated friction model of formula 2# and formula 3# were similarly obtained. The results of fitting the correlation coefficient of the friction model are shown in Table 3.

The coefficients obtained from the fits in Table 3 were substituted into Equation (6) to obtain a thermodynamically coupled integrated friction model for each of the three formulations.
(7)$$\mu =\left(0.2238e^{-0.002231{\left(t-70\right)}^{2}}+1.013\right)\left(10.3850p^{-0.3834}v^{1.200}+0.3975\right)$$
(8)$$\mu =\left(0.6278e^{-0.001974{\left(t-68\right)}^{2}}+0.980\right)\left(12.1476p^{-0.4440}v^{1.550}+0.3204\right)$$
(9)$$\mu =\left(0.8712e^{-0.002744{\left(t-73\right)}^{2}}+1.028\right)\left(71.9234p^{-0.6167}v^{1.455}+0.5443\right)$$
where *t* is the temperature, °C; *p* is the contact pressure, N; and *v* is the sliding speed, mm/s.

Equation (7) is the friction model for formula 1#, Equation (8) is the friction model for formula 2#, and Equation (9) is the friction model for formula 3#.

### 4.2. Validating the Friction Model

#### 4.2.1. Comparison of the Model Predictions and Experimental Test Data under Different Contact Pressures and Temperatures

The experimental data in Appendix A were used to establish a thermodynamically coupled integrated friction model. The experimental data in Appendix B were used to validate the prediction results and accuracy of the model. First, the model prediction was compared with the experimental test data at different contact pressures and temperatures, and the data comparisons are shown in Figure 14.

The prediction model data are shown in Appendix A, and the thermally coupled integrated friction models of the three formulations are shown in Equations (7)–(9); the experimental validation data are shown in Appendix B. The curves are the thermally coupled integrated friction models, and the discrete points are the experimental data, as shown in Figure 14. Under the condition of the same contact pressure, with the increase in the rubber temperature, the friction coefficient gradually increased to the maximum value and then decreased, and under the condition of the same temperature, the increase in the contact pressure reduced the friction coefficient. For formula 1#, the experimental data at 40 °C were slightly higher than the model-predicted value, which may be due to the high friction shear force caused by the high tension of the abrasive belt; the experimental data at 85 °C were slightly lower than the model-predicted value, which may be due to the high temperature conditions of the rubber softening and abrasive debris adhering to the abrasive belt caused by the friction decrease; the agreement between the 55 °C and 70 °C experimental data and the model-predicted values was better. For formula 2#, some experimental data at 40 °C and 55 °C were slightly lower than the model-predicted values, due to the low tension of the abrasive belt, resulting in the friction shear force being too low; for formula 3#, the experimental data under the contact pressure of 300 N had a certain gap with the model-predicted values, and the curve shifted to the right a little bit, due to the wear of the rubber wheel in the process of friction resulting in a smaller outer diameter and the replacement of the rubber wheel. The experimental data under the other contact pressures were in good agreement with the predicted model values. For example, the prediction errors of formula 1# at a temperature of 55 °C were 1.5%, 1.0%, 2.5%, and 2.4%, respectively, and the overall experimental data were not much different from the model-predicted values except for some data, with an error of less than 5%.

#### 4.2.2. Fitting Results of the Model to the Experimental Data at Different Slip Velocities

The model predictions were compared with the experimental test data for different temperature conditions at different slip rates, and the data comparison is shown in Figure 15.

The prediction model data are shown in Appendix A, and the thermally coupled integrated friction models of the three formulations are shown in Equations (7)–(9); the experimental validation data are shown in Appendix B. The curves are the thermally coupled integrated friction models, and the discrete points are the experimental data, as shown in Figure 15. Under the condition of the same slip speed, the friction coefficient gradually reached the maximum value and then decreased with the increase in the rubber temperature, and under the condition of the same temperature, the increase in the slip speed increased the friction coefficient. For formula 1#, there was a certain gap between the experimental data of the slip speed of 100 mm/s and the model-predicted value, the experimental data were slightly higher than the model-predicted value under the condition of 40 °C, and there was not much difference between the experimental data and the model-predicted value under the conditions of 55 °C and 85 °C. For formula 2#, there was a certain gap between the experimental data and the model-predicted value under the condition of 100 mm/s and the model-predicted value under the condition of 55 °C, and the experimental data were slightly lower than the model-predicted value under the condition of 55 °C; for the other values, there was not much difference between the slip speed experimental data and the model-predicted value. For formula 3#, in 70 °C conditions, some of the experimental data were slightly lower than the model-predicted value, which may be due to the high temperature causing rubber softening and abrasive chips adhering to the abrasive belt caused by the fluctuation in the test. For formulas 1#, 2#, 3#, in the slip speed of 100 mm/s conditions, there was a certain gap between the numerical values of the experimental data and the model-predicted values. The main reason is that the friction experiment under the condition of 100 mm/s was run first, which would be directly related to the pre-grinding condition of the rubber wheel, the quality of the vulcanization, the installation, the flatness, out-of-roundness, and other factors, and with the prolongation of the experimental time and the accumulation of wear and tear, the error caused by the above initial factors was gradually reduced. For example, the prediction errors of formula 1# at the 200 mm/s slip speed were 0.8%, 2.1%, 1.1%, and 2.47%, respectively. Except for individual data points, the overall experimental data were not much different from the model prediction values, and the errors were within 5%.

## 5. Conclusions

(1) A fully automatic multi-working condition friction tester has been developed. It can realize the friction performance test of rubber under the interaction of different temperatures, slip rates, loads, frequencies, material formulas, road conditions, etc.

(2) The coefficient of friction of rubber is related to the contact pressure, slip speed, and temperature. The increase in pressure leads to a decrease in the adhesive friction and thus a decrease in the coefficient of friction. An increase in slip velocity leads to an increase in the adhesive friction and thus an increase in the coefficient of friction. The increase in temperature leads to a decrease in the rigidity of the rubber wheels, the adhesion friction increases, and the coefficient of friction shows an increasing trend. After the temperature increases to a certain extent, the growth trend of the adhesion friction slows. The hysteresis loss of the rubber wheel gradually decreases with the increase in the temperature. When the reduction in the hysteresis friction exceeds the increase in the adhesion friction, the friction coefficient shows a decreasing trend. This leads to the coefficient of friction, with the increase in temperature showing a trend of first increasing and then decreasing.

(3) Based on the basic theory of friction, such as the Arrhenius equation, and experimental research, the Dorsch friction model was modified in terms of temperature, and the analytical relationship between the rubber friction coefficient and the combined variables of contact pressure, slip velocity, and temperature was established, which is more in line with the actual situation of rubber friction. The predicted data of the thermodynamic coupling model of rubber friction are compared with the experimental data, and the error is controlled within 5%, which verifies the accuracy of the model and provides a theoretical basis for the study of rubber friction.

## Figures and Tables

**Figure 1 polymers-16-00596-f001:**
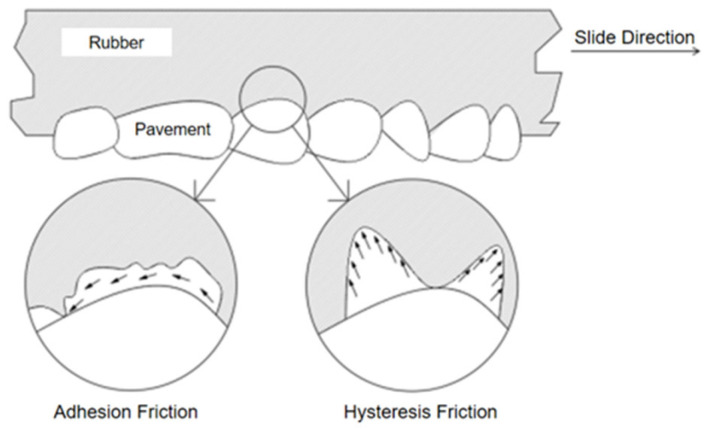
Key mechanisms of tire–pavement friction.

**Figure 2 polymers-16-00596-f002:**
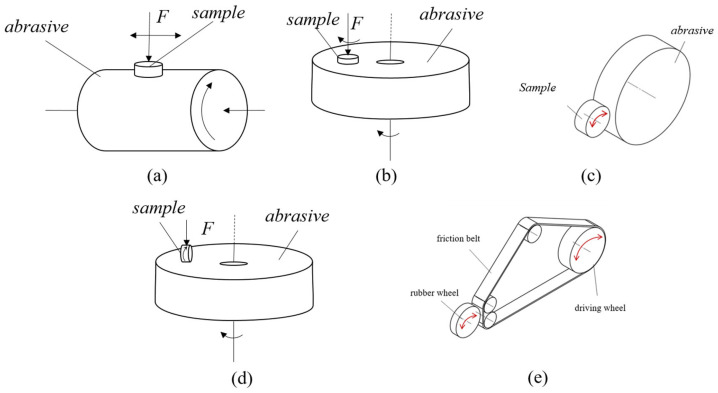
Schematic diagram of the internal structure of the friction chamber. (**a**) Pin cylinder wear test machine, such as a DIN wear test machine. (**b**) Pin disc wear test machine, such as a Williams wear test machine. These two types are mainly used to test the sliding wear behavior of the plane. (**c**) Rotary friction and wear test machine, which can be used for testing products such as tires, such as a Lambourn wear test machine and an Akron wear test machine. (**d**) Test machines that combine rotation and sliding methods, such as a LAT100 abrasion test machine. (**e**) Contact form of the fully automatic multi-working condition friction testing machine developed in this research.

**Figure 3 polymers-16-00596-f003:**
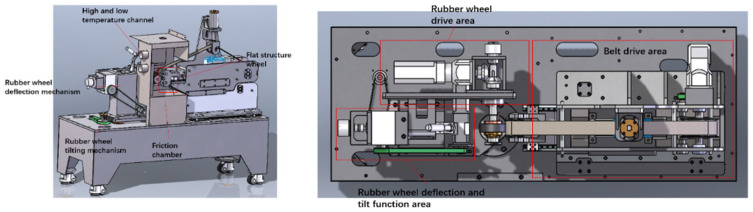
Schematic diagram of the functional area of the main machine of the fully automatic multi-working condition friction tester.

**Figure 4 polymers-16-00596-f004:**
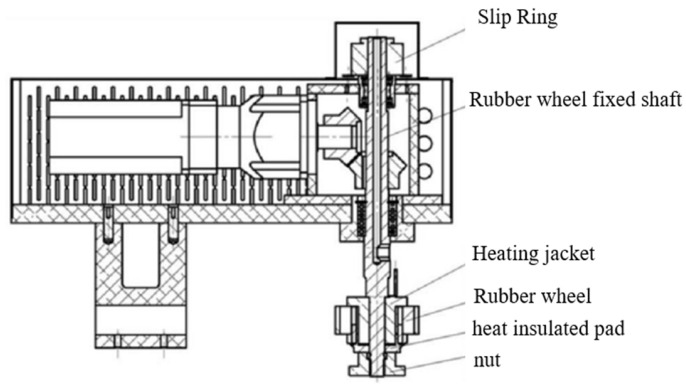
Top view of the partial structure of the rubber wheel and heating system.

**Figure 5 polymers-16-00596-f005:**
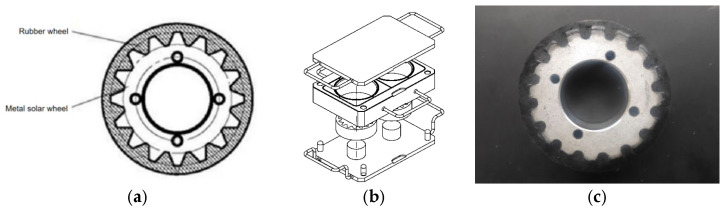
Schematic diagram of the rubber wheel structure and vulcanization molds. (**a**) Structure diagram of the rubber wheel. (**b**) Schematic diagram of the vulcanization mold. (**c**) Physical picture of the rubber wheel.

**Figure 6 polymers-16-00596-f006:**
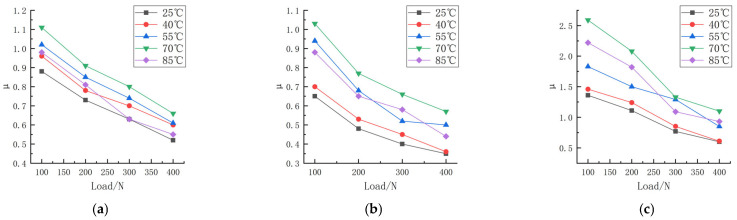
Friction coefficient of three formulations at different pressures and temperatures. (**a**) Formula 1. (**b**) Formula 2. (**c**) Formula 3.

**Figure 7 polymers-16-00596-f007:**
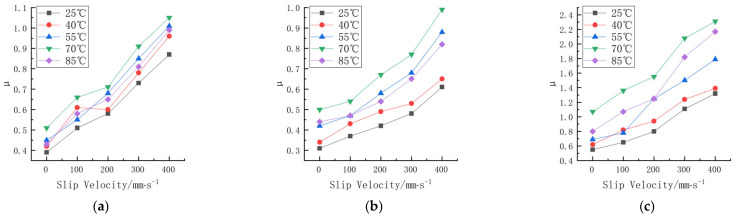
Friction coefficients of three formulations at different slip speeds and temperatures. (**a**) Formula 1. (**b**) Formula 2. (**c**) Formula 3.

**Figure 8 polymers-16-00596-f008:**
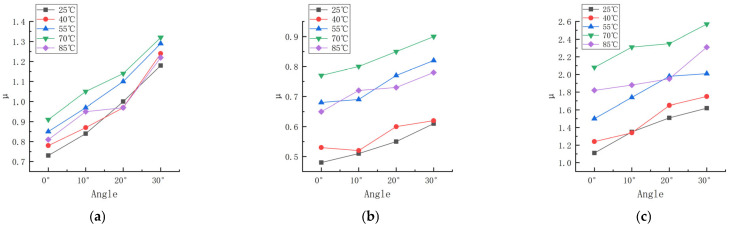
Friction coefficients of three formulations at different deflection angles and temperatures. (**a**) Formula 1. (**b**) Formula 2. (**c**) Formula 3.

**Figure 9 polymers-16-00596-f009:**
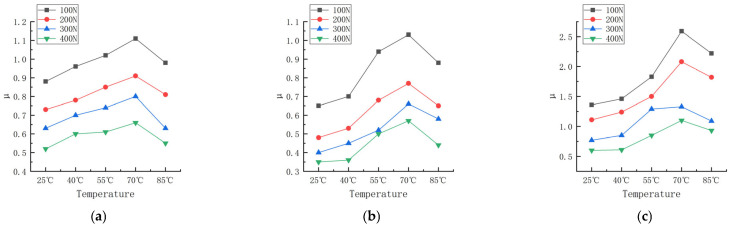
Friction coefficients of the three formulations under different pressure and temperature conditions. (**a**) Formula 1. (**b**) Formula 2. (**c**) Formula 3.

**Figure 10 polymers-16-00596-f010:**
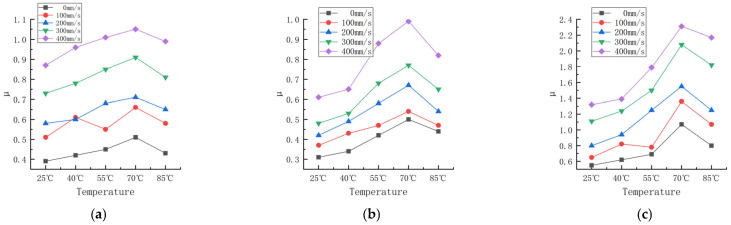
Friction coefficients of the three formulations under different slip speeds and temperatures. (**a**) Formula 1. (**b**) Formula 2. (**c**) Formula 3.

**Figure 11 polymers-16-00596-f011:**
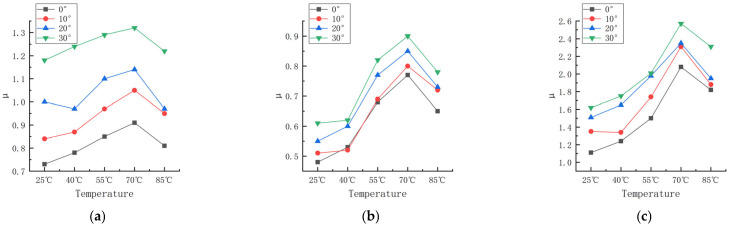
Friction coefficients of the three formulations under different deflection angles and temperatures. (**a**) Formula 1. (**b**) Formula 2. (**c**) Formula 3.

**Figure 12 polymers-16-00596-f012:**
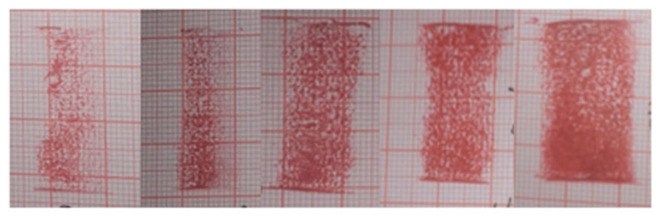
The rubber wheel impressions at different temperatures under the same load, from left to right: 25 °C, 40 °C, 55 °C, 70 °C, and 85 °C.

**Figure 13 polymers-16-00596-f013:**
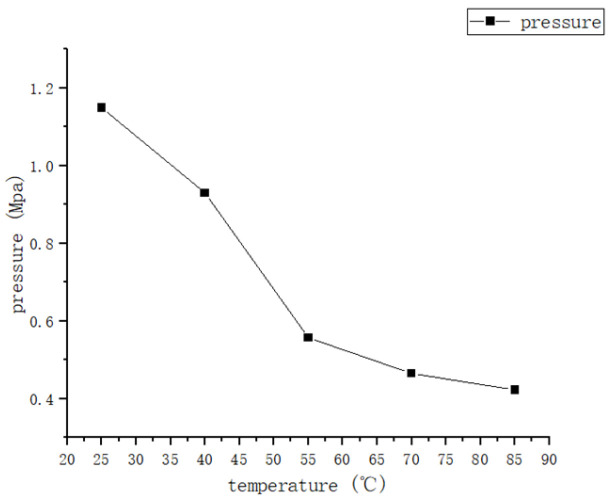
The relationship between pressure and temperature.

**Figure 14 polymers-16-00596-f014:**
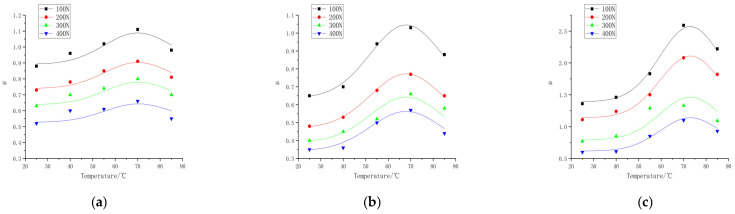
Predicted curves of the thermally coupled integrated friction model versus data at discrete points of the experimental test data (Comparison of the model predictions and experimental results for three formulations at temperatures of 25, 40, 55, 70, and 85 °C and pressures of 100, 200, 300, and 400 N. Curves are model predictions, and discrete points are experimental test results). (**a**) Formula 1. (**b**) Formula 2. (**c**) Formula 3.

**Figure 15 polymers-16-00596-f015:**
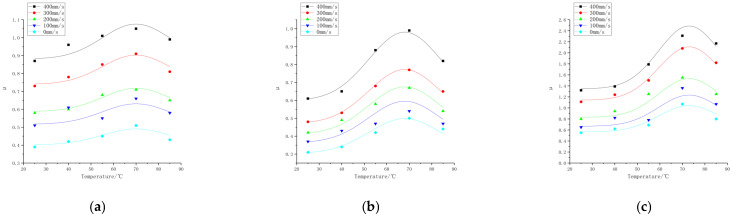
Predicted curves of the thermally coupled integrated friction model versus data at discrete points of the experimental test data (Comparison of the model predictions and the experimental results for three formulations at temperatures of 25, 40, 55, 70, and 85 °C and slip velocities of 0, 100, 200, 300, and 400 mm/s. Curves are model predictions, and discrete points are experimental test results). (**a**) Formula 1. (**b**) Formula 2. (**c**) Formula 3.

**Table 1 polymers-16-00596-t001:** Composition and proportion of the three rubber formulations.

Formula 1#		Formula 2#		Formula 3#	
Component	phr	Component	phr	Component	phr
NR	100.00	3#RSS	100.00	STR20	70.00
N234	60.00	N115	53.00	BR	30.00
Stearic acid	1.00	175GR	7.00	N234	55.00
ZnO	3.50	ZnO	3.50	TXN-203	2.00
Ceresine wax	1.50	Stearic acid	2.00	ZnO	3.50
4020	1.50	Ceresine wax	1.00	Stearic acid	2.00
RD	1.50	4020	2.00	Ceresine wax	1.00
S	1.10	RD	1.50	4020	2.00
NS	1.20	Resin	5.00	RD	1.00
		S	1.20	NS	1.00
		CZ	1.10	S	1.40
				CTP	0.17

**Table 2 polymers-16-00596-t002:** Experimental conditions.

Slip Velocity /mm·s^−1^	Load /N	Sideslip Angle /°	Temperature /°C
0	100	0	25
100	200	10	40
200	300	20	55
300	400	30	70
400			85

**Table 3 polymers-16-00596-t003:** Correlation coefficient fitting results of friction model.

Fitting Parameters	Compound Formulations		
	Formula 1#	Formula 2#	Formula 3#
*α*	0.2238	0.6278	0.8712
*β*	0.002231	0.001974	0.002744
*γ*	1.013	0.980	1.028
*a* _1_	10.3850	12.1476	71.9234
*a* _2_	0.3975	0.3204	0.5443
*b* _1_	−0.3834	−0.4440	−0.6167
*b* _2_	1.200	1.550	1.455
*t_m_*	70	68	73

## Data Availability

The authors confirm that the data supporting the findings of this study are available within the article.

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
