# Peer review of "A Study of the Friction Characteristics of Rubber Thermo-Mechanical Coupling"

_polymers, 2024, doi:10.3390/polym16050596_

Round 1
Reviewer 1 Report
Comments and Suggestions for Authors
Title: Study on the Friction Characteristics of Rubber Thermo-Mechanical Coupling
Manuscript ID: polymers-2822847
Recommendation and Comments
In this manuscript, the authors did an experiment to develop a fully automatic multi-condition friction tester to study the correlation between rubber friction characteristics and various factors such as speed, pressure, temperature, side deflection angle, and side inclination angle and to establish the characterization of rubber friction on multiple factors. In addition, the authors have found that understanding the friction mechanism between tires and road surfaces is indeed complex and pivotal for accurately characterizing the friction coefficient under varying factors, and existing research lacks comprehensive insights into the intricate thermal coupling characteristics impacting rubber friction.
English writing in the manuscript should be revised. Some details in this manuscript need to be corrected with citing. I suggest that the manuscript needs major revision.
1. In section 1, there are a total of 9 citations in the paper. 7 of all citations are in the introduction. The citations should be enriched to understand and compromise the study. Especially, it should be cited in the last 5 years studies.
2. In section 2, the items of subtitles in section 2 should be corrected. No 1.2 item in the section 2.
3. In section 2, Figure 2 should be removed from the paper because it seems like device advertisement. Figure 4 is sufficient to explain the multi-condition friction tester.
4. In section 2, Figure 3 should be clarified in the caption at the figure. For instance, what should we do understand in the figure 3(b)?
5. In section 2, the caption of the figure in figure 6 should be revised.
6. In section 2, why did the authors need figure 7? The friction test parameters should be informed with table or text in the paper.
7. In section 2, “selection of experimental conditions” should be explained with table to visibility of the paper.
8. In section 3, there are no citations in section 3. The study should be supported with citations.
9. In section 3, “…due to the fact that the increase in pressure when the rubber is rubbed on a rough surface leads to a decrease in the adhesive friction thus making the coefficient of friction decrease…” could you give the reference this sentence.
10. In section 3, “…the difference in temperature will lead to a large change…” English text should be checked and revised. Same issue: “…which will not be repeated because other road surface characteristics are not selected in this paper.”
11. In section 3, the pixels of the figures should be increased to visibility.
12. In section 3, the explanations in the paper are not sufficient due to not including the references.

Reviewer 2 Report
Comments and Suggestions for Authors
The topic of the paper is interesting. However, the presentation of the paper is not good. My recommendation is a reject. The manuscript is more like a report than a paper. Below are my comments:
1. All of the mentioned citations, including those listed in the Reference section, are currently outdated. This makes the paper lose its state of the art. Besides, the paper's originality is questionable. Please include a minimum of 10 recent sources of literature.
2. The manuscript contains numerous typographical errors (line 358, 368, 379, etc).
3. The conclusion is too long and is not supported by the data.
4. One basic thing that needs to be underlined is that there is a contradiction in the methods used. On the one hand, the author claims that he will derive an analytical equation (line 519), namely the thermal coupling model of rubber friction. But on the other hand, the author uses the Dorsch friction equation. With this approach, the author actually only carried out curve fitting from the results of this experiment. This makes the paper less interesting from an academic perspective
Reviewer 3 Report
Comments and Suggestions for Authors
It would be helpful to develop a friction tester of rubber for various conditions. The requirement would be achieved by using the developed friction tester in this manuscript.
While the manuscript is well written, I cannot accept the manuscript as is since I still have some questions about this manuscript.
My major comments area as follows:
1) Fig. 1: for hysteresis friction, why are there two peaks of pressure? Please add explanation about it to the manuscript.
2) Page 2, line 80: I guess the range of glass-transition temperature among three rubber compounds is -70--100 degree Celsius and too small to universally discuss the reasonability of the friction tester.
3) Page 3, line 90: please add some explanation about the friction tester to Fig. 2. It is difficult to understand. In addition, was each test conducted by using the same rubber specimen? I guess the friction test results would depends on the order of friction test for each condition.
4) Page 13 Table 2: I cannot understand the meaning of each parameter. I guess each parameter would be determined for each rubber composition by fitting. However, the relationship between each parameter and physical properties of rubber.
Round 2
Reviewer 2 Report
Comments and Suggestions for Authors
The authors have incorporated the suggestions given by the reviewer and addressed the issues raised. Also the manuscript has now been revised thoroughly which improved the expression of the results. Therefore I recommend the publication of this paper in this journal.
Author Response
Thank you for your comments on this paper and good luck with your work.
Reviewer 3 Report
Comments and Suggestions for Authors
All issues have been properly addressed.
Author Response

(The authors gave the same response as above.)
